# Green Product Types Modulate Green Consumption in the Gain and Loss Framings: An Event-Related Potential Study

**DOI:** 10.3390/ijerph191710746

**Published:** 2022-08-29

**Authors:** Guanfei Zhang, Mei Li, Jin Li, Min Tan, Huie Li, Yiping Zhong

**Affiliations:** 1Department of Psychology, School of Education Science, Hunan Normal University, Changsha 410081, China; 2Cognition and Human Behavior Key Laboratory of Hunan Province, Changsha 410081, China

**Keywords:** green product type, green consumption behavior, gain framing, loss framing, event-related potential (ERP), P3, P260

## Abstract

People show a separation of intention and behavior in green consumption, and promoting actual green purchase behavior is more important than purchase intention. This study adopted a conflicting environmental decision paradigm to investigate behavioral and neural processes during actual green consumption decision-making involving different types of green products and message framing, according to construal level theory. Participants were instructed to make green consumption decisions involving green products with different psychological distances (self-interested green products vs. other-interested green products) under gain (e.g., buying green products brings positive results) or loss framing (e.g., not buying green products brings negative effects) while electroencephalograms were recorded. The behavioral results demonstrated that participants tended to purchase green products under loss framing more than under gain framing. The event-related potential results showed that under gain framing, decision-making for self-interested green products was associated with larger P3 than decision-making for other-interested green products. While under loss framing, decision-making for other-interested green products has a larger P260 than for self-interested green products. These findings suggest that under gain framing, self-interested green products elicit more cognitive resources than other-interested green products, while under loss framing, other-interested green products elicit stronger negative emotions than self-interested green products. The research has managerial implications for promoting consumers’ actual purchase behavior.

## 1. Introduction

With the promotion of environmental protection initiatives around the world, people have begun paying attention to the future of the earth and their own quality of life. Green products have also attracted consumers’ attention because of their environmentally friendly attributes of saving resources and energy and their health attributes, such as the absence of artificial substances [1]. This germination of environmental awareness makes green consumption possible [2,3,4,5]. Green consumption is a behavior satisfying human needs or desires with the least harmful impact on the natural environment [6]. As an important measure to achieve carbon peaking and carbon neutrality, it has received extensive attention from scholars in the fields of environmental studies, psychology, and marketing [7]. The essence of green consumption behavior is a trade-off between self-interest and environmental interest because it requires individuals to use their own funds (self-interest) to purchase green products and contribute to environmental friendliness (environmental interest) [8]. Therefore, although many consumers show a high willingness to purchase green products, their actual purchase behavior is low, that is, they show an “attitude–behavior gap” [9,10,11]. Previous studies have focused more on marketing methods to promote individual green purchase intention [12,13,14] or marketing methods that claim to be focused on promoting green purchase behavior, but in fact, the dependent variable still measures verbally reported willingness to buy green products at no cost to the individual (e.g., “I’ll buy green products even if they are more expensive than other products”) [15,16]. These studies ignored the existence of this “attitude–behavior gap”. Therefore, it is very important to find marketing solutions that can increase the actual purchase behavior of the public towards green products.

Previous studies have shown that green consumption behaviors are affected by the types of green products, classified according to their product attributes and consumer purchasing motivations [1,17]. Green products have two types: self-interested and other-interested [1,18]. Self-interested green products are products for which consumers make purchasing decisions based on self-interested motives, such as health concerns or cost savings (e.g., organic fruits or vegetables). For other-interested green products, consumers make purchasing decisions based on altruistic motives to improve environmental resources (e.g., degradable garbage bags) (see Table 1 for more variable information) [17]. Previous studies have reached no consistent conclusion about which type of green product induces higher purchase intention. For example, Peattie [19] found that green products that can bring future benefits to an entire generation generate higher purchase intentions than those that can only bring individual benefits. This is because of the role played by the norms of environmental protection advocated in society. Pro-environmental behaviors with purely altruistic motives receive more societal approval and are more acceptable under environmental norms than those with self-interested motives [20]. However, Liu, Li and Song [17] did not find this effect, instead finding that the two types are equivalent in terms of green consumption. Since different types of green products will affect consumers’ green purchasing decisions, it is necessary to consider the marketing measures for different types of green products when exploring the marketing plan for green products. At the same time, we found that these studies focused on measuring consumers’ green purchase intention or attitude, but, as mentioned above [9,10,11], there is a gap between intention and behavior, and, more importantly, research should focus on actual green purchase behavior. Therefore, it is worth paying attention to what kind of marketing plan is adopted for different types of green products to promote actual green purchase behavior.

Previous studies have found that different types of green products have different psychological distances [6,22,23]. Psychological distance is a subjective experience, referring to the proximity or distance of a stimulus from the self, the location of the self, and the time point of the self [24]. Other-interested green products have a farther psychological distance from individuals because the direct beneficiaries are mainly related to the environment rather than the individuals; self-interested green products are related to the individuals’ own interests, so they are less psychologically distant [6]. Importantly, Kulkarni and Yuan [25] found that the role of psychological distance in the persuasive effect of product advertising is moderated by message framing. Specifically, in gain framing (that is, buying products brings positive results), far psychological distance produces better advertising persuasion effects than close psychological distance, while, in loss framing (that is, not buying products brings negative effects), manipulation of near and far psychological distance produced the same advertising persuasion effects. Therefore, message framing may be effective in promoting the green purchasing behavior of different types of green products.

Construal-level theory, from the perspective of social cognition, proposes that people’s reactions to things depend on their mental representations of them, and mental representations have different levels that depend on the degree of psychological distance [23,26,27]. When individuals perceive a greater psychological distance, they tend to use abstract features that reflect the essence to characterize things and focus on desirability (that is, the final result’s state of the behavior, reflecting the “why” of doing the behavior). Conversely, when perceiving less psychological distance, they tend to use non-essential, specific characteristics to characterize things, focusing on feasibility (that is, the means or difficulty of reaching the behavioral result state, reflecting the “how” to do this behavior) [23,27,28]. Gain framing emphasizes that the purchase of green products will lead to positive and desirable benefits. Such positive events and states prompt people to consider the reasons and values of green purchases from an abstract long-term perspective. Therefore, under gain framing, other-interested green products, such as far-distance green products, are more able to promote consumers’ green purchasing behavior than close self-interested green products. Loss framing emphasizes the negative and non-desirable loss caused by not purchasing green products. This state of loss not only makes people pay attention to what actions should be taken to avoid losses at close distance [28,29], but also, according to the principle of loss aversion, such negative events and states will cause the transfer of individual attention [30], focusing more on the loss itself and ignoring the effects of psychological distance [31]. Therefore, under loss framing, the green purchasing behaviors of individuals for self-interested green products with close psychological distance and other-interested green products with long psychological distance are similar. Therefore, we propose the following hypotheses: 

**Hypotheses** **1a** **(H1a).**
*Under gain framing, participants would have a higher proportion of green purchases for other-interested green products than for self-interested green products.*


**Hypotheses** **1b** **(H1b).**
*Under loss framing, there would be no significant difference in the proportion of green purchases of other-interested green products and self-interested green products.*


Event-related potential (ERP) is a neuroscience technique with high temporal resolution that can be used to assess the temporal course of brain activities in social decision-making and the underlying neural mechanisms [32]. Most of the previous studies focused on the behavioral experimental level of green consumption, focusing on green consumption willingness or behavior, but fewer explored the neural mechanism [1,6,29,33,34]. However, many neuromarketing researchers have proposed that it is of great significance to explore the neural mechanisms behind green consumption behavior to promote the marketing of green products [35,36]. Hence, we conducted an ERP study to examine how green consumption behaviors under gain and loss framings were affected by green product type from a neurological perspective (see Table 2 about the related research on neuromarketing integration). Previous ERP studies on green consumption and social decision-making have mainly examined two components of ERP: P260 and P3 [37,38,39,40]. P260 is a positive visual component that peaks during 200–300 ms after stimulus presentation, while considered a dilemma-induced conflict and social emotion [41,42]. Previous studies have found that individuals facing a dilemma concerning saving people in an earthquake have a larger P260 when the people to be rescued are their own parents than when they are two strangers. P3 is a positive component that peaks between 300–500 ms after stimulus onset and represents the late conscious stage of cognitive processing [39,43]. The P3 component is often related to the allocation of attention and cognitive resources in decision-making [39,44] and represents stimulus-induced motivation levels [45]. For example, compared with stimuli with low self-relevance (e.g., the name of a foreign country leader), a stimulus with high self-relevance (e.g., one’s own name [46]) evokes a larger P3 amplitude. This is because self-related stimuli elicit higher levels of motivation than other stimuli, and, thus, more cognitive resources are invested [47].

As outlined in the introduction section, we predicted that, under gain or loss framing, green products with different psychological distances would elicit different neural responses to green consumption behavior, indexed by P260 and P3. First, the altruistic motives of other-interested green products are more socially acceptable and more in line with environmental-protection standards than the egoistic motives of self-interested green products [20], and the description of not purchasing other-interested green products under loss framing leads to a greater sense of conflict and negative emotions than not purchasing self-interested green products. P260 is associated with greater dilemma conflict as well as negative emotions [42]. We therefore expected that:

**Hypotheses** **2** **(H2).**
*Under loss framing, other-interested green products would induce greater P260 than self-interested green products.*


Second, self-interested green products elicit higher motivation levels because of their psychological nearness to individuals [6], and enhanced P3 is associated with greater motivation [49]. We therefore expected an increase in P3 when people decide to buy a self- (vs. other-) interested green product under gain framing. In contrast, under loss framing, according to the theory of loss aversion, losses have a larger impact on decision-making than equal gains [30]. Due to the strong influence of loss on decision-making, loss will affect the individual’s focus, causing the individual to focus on the loss itself and ignore the focus on the near and far psychological distance [31]. Therefore, we expected that, under loss framing, there would be no psychological distance effect of green product type on P3. The hypotheses about P3 are as follows:

**Hypotheses** **3a** **(H3a).**
*Under gain framing, the P3 amplitude induced by self-interested green products would be significantly higher than that for other-interested green products.*


**Hypotheses** **3b** **(H3b).**
*Under loss framing, there would be no significant difference in the P3 amplitudes induced by self-interested green products and other-interested green products.*


We drew a technical roadmap to enhance the academic logic of the article, as shown in Figure 1.

## 2. Materials and Methods

### 2.1. Participants and Experimental Design

A power analysis (G*Power 3.1, power = 0.90, effect size f = 0.25, α = 0.05) suggested that 30 participants would ensure 90% statistical power in the case of small to medium effect sizes [50]. Thirty-five undergraduate or graduate students (21 female and 14 male; aged between 18 and 25, mean 20.43 years) from Hunan Normal University participated in this experiment, which had a 2 (green product type: self-interested vs. other-interested) × 2 (message framing: gain vs. loss) within-participants factorial design [32,33,37,39,51]. All participants were right-handed, had normal or corrected-to-normal vision, had no history of mental illness, and signed informed consent before the experiment. According to the purchase decision of the participants in the experimental task, a certain reward would be given after the experiment. This study was approved by the Ethics Committee of the Department of Psychology, Hunan Normal University (No.: 2022-343).

### 2.2. Experimental Materials

#### 2.2.1. Green Product Type Manipulation

To ensure the validity of the experimental materials, green product types were classified based on the definition in the literature [1,17] and Baidu (https://www.baidu.com/, access on 20 May 2021) and Bing (https://cn.bing.com/, access on 20 May 2021). Eighteen kinds of green products were collected, including nine kinds of self-interested green products (e.g., organic apples) and nine kinds of other-interested green products (e.g., degradable garbage bags). We conducted a preliminary experiment to test the types green products belong to, the familiarity, arousal, and product preference. The green products we chose could well represent the type they belong to, and there was no significant difference in these three attributes between the two types of green products (see Appendix A).

#### 2.2.2. Message Framing Manipulation

The manipulation of the message framing was adapted from Baek and Yoon [33] and Zubair, Iqbal, Usman, Awais, Wang and Wang [37] The gain framing manipulation was “Buy green products, and the environment (you) obtains water and soil (health)”, and the loss framing manipulation was “Do not buy green products, and the environment (you) loses soil and water (health)”. The effectiveness of the message framing manipulation was tested (see Appendix A). Results suggested that both manipulations were effective and could successfully induce the participants’ feelings of gain and loss. Those who participated in the experimental evaluation did not participate in the formal experiment.

#### 2.2.3. The Dilemma of the Environmental Decision Paradigm

We used the dilemma of the environmental decision paradigm in this experiment [49,52,53]. The names and prices of common (left) and green products (right) appeared at the same time, and participants were required to make purchase choices as quickly as possible. The task was as follows: The left side of the interface contained the name and price of common products; the right side contained the name and price of green products [49,53]. Since the essence of green consumption behavior is the balance between self-interest and environmental interest [8], we have set up a dilemma conflict in this purchase decision. We gave the participants ¥55 to purchase products and informed them that they would choose which products to buy in many rounds. After the experiment, we would randomly select one of the many products they chose and deduct the cost of the corresponding product (common or green product) from the payment for participation [52,53]. Thus, participants who purchased green products would receive less payment than those who purchased common products because the price of green products was always higher. This resulted in a dilemma conflict—that is, whether the participants wanted to buy common products to get more payments (i.e., good for personal interests) or green products to get less payments (i.e., good for environmental benefits).

A study found that the green product premium that consumers can accept is 10% higher than that of common products, and when it is higher than 10%, consumers’ green purchasing behavior will be greatly reduced [54]. However, according to Alibaba’s survey, the price of green products generally has a premium of about 33% [55]. Therefore, to ensure the ecological validity of the price of green products in the experiment and to avoid the extreme situation in which participants always choose to purchase green products when their price is low, we followed the research paradigm of Jung, Sul, Lee, and Kim [52] to manipulate the price level, that is, set the price of green products as 25%, 50%, 75%, 100%, 125%, 150%, and 175% higher than that of common products. The price level in the paradigm did not affect our main variables (see Appendix A).

### 2.3. Experimental Procedure

During the purchase task, participants were seated comfortably in a chair with their eyes approximately 75 cm from the computer monitor. As illustrated in Figure 2, in each trial, participants would first see a “+” fixation point for 1000 ms to indicate that the experiment had started, followed by a picture and name of a green product for 1500 ms. At random intervals of 800–1200 ms after that, the manipulation sentence of the message framing was presented for 2000 ms. Then, after a random interval of 1000–1200 ms, the purchase decision interface was presented for 5000 ms. The next trial began after 1000 ms. The total number of trials was 252 with 63 trials in each of the 4 experimental conditions, and the presentation order was pseudo-random based on the type of message framing. The whole experiment was divided into four blocks, and the participants were given sufficient rest between blocks. The experimental task was to carefully watch the green products presented in the experiment. This included clarifying what green products they would buy and the type they belonged to. Participants then carefully read the statements in the subsequent message framing interface and then made a choice as soon as possible in the purchase decision interface according to their feelings. The decision-making method was a button response; the experimenter told the participants to press the “F” key if they decided to buy a common product and the “J” key if they wanted to buy a green product. The participants were required to respond within 5000 ms, and more than 5000 ms was regarded as unresponsive. Before the formal experiment, the participants conducted a short practice experiment to familiarize themselves with the experimental procedure. Information on the green product materials, including the name, characteristics, and type, appeared before the formal experiment started to give the participants enough time to become familiar.

### 2.4. Electroencephalogram (EEG) Recording and Analysis

The ANT Neuro system was used to record EEG signals using a 64-lead electrode cap extended by the international 10–20 system. During EEG recording, CPz was used as the online reference electrode, and data were re-referenced to the average of left and right mastoid electrodes. The filter bandpass of the online recording is 0.01~100 Hz, the sampling rate of the signal recording is 500 Hz, and the impedance between all electrodes and the scalp is less than 5 kΩ. Offline analysis of the EEG data was performed using MATLAB 2014a and the EEGLAB toolbox (v.13.4.4 b, Swartz Center for Computational Neuroscience, La Jolla, CA, USA). The EEG data were low-pass filtered below 30 Hz and segmented into epochs from 200 ms before and until 800 ms after the onset of the participants’ decision button. After baseline correction (−200 to 0 ms), epochs containing artifacts exceeding ±75 μV were excluded from further analysis. Subsequently, the epochs were averaged separately for each condition of each participant.

Two ERP components, P260 and P3, were analyzed. The P260 and P3 were measured as the mean amplitude within the time window of 210–280 ms and 315–415 ms, respectively. According to the topographical distribution and previous studies suggesting that P3 is a central-parietal component [38,39], the P3 was calculated across nine electrode sites in the central-parietal region (CP3, CPz, CP4, P3, Pz, P4, PO3, POz, and PO4). Based on the topographical distribution of each ERP component and previous literature [40,42,56], P260 statistics were reported across 10 electrode sites in the central-parietal region (CP1, CP3, CPz, CP2, CP4, P1, P3, Pz, P2, and P4). Mean amplitude values were averaged for all selected electrode sites. All data were statistically analyzed using SPSS 26.0. The P260 and P3 amplitudes were analyzed using a two-way repeated-measures analysis of variance (ANOVA) of 2 (green product types: self-interested vs. other-interested) × 2 (message framing: gain vs. loss). The significance level for all analyses was set at 0.05. Post hoc comparisons were Bonferroni-corrected at *p* < 0.05. The Greenhouse–Geisser correction was conducted to account for sphericity violations whenever appropriate, and the partial eta-squared (ηp2) was reported as a measure of effect size.

## 3. Results

### 3.1. Behavioral Results

See Table 3 and Figure 3 for the total proportion of green product purchases under the four conditions (total proportion of green product purchases = number of choices to buy green products/total number of decisions). We first examined the total proportion of purchasing green products using a repeated-measures ANOVA. There was a significant difference in message framing, *F* (1, 34) = 4.39, *p* = 0.044, η_p_^2^ = 0.11, indicating that the participants preferred to purchase green products in the loss framing (0.43 ± 0.05) more than in the gain framing (0.37 ± 0.04). There was no significant difference for green product type, *F* (1, 34) = 0.30, *p* = 0.586, η_p_^2^ = 0.01. The interaction of green product type with the message framing was not significant, *F* (1, 34) = 0.61, *p* = 0.439, η_p_^2^ = 0.02. 

Finally, we divided the trials into two response categories to examine the decision times based on participants’ product purchases, since the proportion of green product purchases was significantly influenced by message framing (see Table 4). A repeated measure analysis of variance (ANOVA) of 2 (purchase decision response: common products vs. green products) × 2 (green product type: self-interested vs. other-interested) × 2 (message framing: gain vs. loss) was conducted for the decision times. There was a significant difference in purchase decision response, *F* (1, 34) = 7.12, *p* = 0.012, η_p_^2^ = 0.17, indicating that the participants spent more time in deciding to purchase green products (864.74 ± 57.59) than common products (769.78 ± 55.63).

Importantly, the interaction between purchase decision response and green product type was marginally significant, *F* (1, 34) = 3.79, *p* = 0.060, η_p_^2^ = 0.10. A simple effects test indicated that when making purchase decisions for self-interested green products, the decision to purchase green products (877.95 ± 56.36) took more time than common products (744.23 ± 60.71), *F* (1, 34) = 9.46, *p* = 0.004. However, when making a purchase decision for other-interested green products, there was no significant difference in the decision-making time required for the decision to purchase common products (795.34 ± 57.64) and green products (851.52 ± 60.43), *F* (1, 34) = 2.20, *p* = 0.147. No other significant effects were observed (*ps* > 0.05) (see Figure 4).

### 3.2. ERP Results

#### 3.2.1. P260 (210–280 ms)

A repeated-measures ANOVA of P260 amplitude revealed a non-significant main effect of green product type, *F* (1, 34) = 0.19, *p* = 0.664, η_p_^2^ = 0.01, and a non-significant main effect of message framing, *F* (1, 34) = 0.01, *p* = 0.930, η_p_^2^ = 0.00. However, green product type × message framing demonstrated a significant interaction effect, *F* (1, 34) = 5.28, *p* = 0.028, η_p_^2^ = 0.13. We conducted a simple effect analysis to investigate this interaction. The results showed that the P260 was more positive for other-interested green products (4.93 ± 0.60 μV) than for self-interested green products (4.31 ± 0.53 μV) in the context of the loss framing, *F* (1, 34) = 7.11, *p* < 0.05, but the P260 showed no difference between self- (4.81 ± 0.54 μV) and other-interested green products (4.38 ± 0.60 μV) in the context of the gain framing, *F* (1, 34) = 1.16, *p* = 0.288 (see Figure 5).

#### 3.2.2. P3 (315–415 ms)

A repeated-measures ANOVA of P3 amplitude revealed a non-significant main effects of green product type, *F* (1, 34) = 0.71, *p* = 0.405, η_p_^2^ = 0.02, and message framing, *F* (1, 34) = 0.91, *p* = 0.346, η_p_^2^ = 0.03. More importantly, green product type × message framing demonstrated a significant interaction effect, *F* (1, 34) = 6.99, *p* = 0.012, η_p_^2^ = 0.17. We conducted a simple effect analysis to investigate this interaction. The P3 was more positive for self-interested (5.74 ± 2.78 μV) than for other-interested green products (5.06 ± 2.93 μV) in the context of the gain framing, *F* (1, 34) = 5.21, *p* < 0.05, but the P3 showed no difference between self- (5.02 ± 2.60 μV) and other-interested green products (5.28 ± 2.61 μV) in the context of the loss framing, *F* (1, 34) = 0.76, *p* = 0.389 (see Figure 5).

## 4. Discussion

The present study examined how a green product type regulated green consumption under a gain and loss framing using a neurophysiological approach. The behavioral results showed that loss framing could increase the purchase ratio of green products more than gain framing. Purchasing green products requires a longer decision-making time than purchasing common products and is particularly prominent for self- (vs. other-) interested green products. In line with our predictions, the ERPs’ results demonstrated that green product type moderated green consumption in the gain and loss framings, as reflected by the mean amplitudes of P260 and P3.

Consistent with our prediction H2, we observed that the P260 amplitude was larger for other-interested green products than for self-interested green products under loss framing. Numerous studies have demonstrated that P260 is related to negative emotions and dilemma conflicts in dilemma decision-making [42,56,57]. This suggested that, under loss framing, other-interested green products are more likely to induce dilemma conflict and negativity than self-interested green products. This is consistent with Zhan, Xiao, Tan, and Zhong [40]’s study, which found that when participants faced the dilemma of shocking subjects of varying social distances to obtain money, a greater P260 was induced when the shocking object was a stranger than when it was the self or a friend. The researchers interpreted this to mean that in this conflict situation of harming others for self-benefit, harming strangers results in greater moral conflict and harm aversion than harming friends and the self. Similar to the mechanism of this study, it used an electric shock task to construct a conflict situation between harming others and self-money gain, and this dilemma in our study is reflected in the conflict between the harm caused by not purchasing environmentally friendly products under loss framing and the monetary benefits saved by common products compared with green products. In our research, we manipulated loss framing so that the green consumption decision-making process would be under the loss situation of “not buying environmentally friendly products will bring negative harm”, where such “harm” includes damage to one’s own health or environmental water and soil resources. Socially responsible consumption is socially oriented rather than self-centered [58]. Therefore, compared with self-interested green products, the purchasing motivation of other-interested green products is more socially recognized, praised, and in line with environmental protection standards [20]. Therefore, the description of other-interested green products under loss framing (i.e., not buying other-interested green products will harm the environment) was contrary to the norms of environmental protection advocated by contemporary society. Thus, more conflict, loss aversion, and negative emotions were generated by purchasing other-interested green products compared to self-interested green products. Other-interested green products induced a larger P260 than self-interested green products when making green consumption decisions under loss framing. Furthermore, under gain framing, it is emphasized that purchasing green products would bring benefits which would not induce disgust or negative emotions in the participants. Therefore, it is reasonable for the participants to have the same degree of P260 under different types of green products.

Consistent with our predictions H3a and H3b, P3 was only sensitive to green products with different psychological distances under gain framing, and the green consumption decision for self-interested green products caused a larger P3 than for other-interested green products. Studies have shown that P3 is associated with higher-level cognitive decision-making processes and represents attention to motivationally significant stimuli and the investment of cognitive resources in this decision [39,44]. Motivated stimuli will preferentially capture resources and lead to larger P3 [45]; for example, stimuli with higher self-relevance will induce larger P3 amplitudes [46]. Previous studies have found that in altruistic decision-making, in a gain context, there is a stronger motivational attention to friends who are psychologically closer than to strangers who are psychologically distant, which then induces a larger P3 [39]. Consistent with these research results, our findings showed that, under gain framing, there was a higher level of motivation for self-interested green products, suggesting that self-interested green products had more significant motivational significance because of the smaller psychological distance between them and the self [6], which induced a larger P3. This also validates construal-level theory [24]. Previous research has found that individuals with low levels of construal experienced greater mental load when dealing with abstract tasks [59]. That is, the mismatch of the construal level will prompt the individual to invest more cognitive resources to process the task, manifesting as an increase in P3. Therefore, other-interested green products are at a high construal level, consistent with gain framing’s construal levels, focusing on the causes and values of far-distance psychological distance. Therefore, compared with the inconsistent construal level of self-interested green products, fewer cognitive resources will be invested; that is, under gain framing, the P3 fluctuation induced by other-interested green products is smaller than that for self-interested green products. On the other hand, from the perspective of environmental norms, under gain framing, the motivation for other-interested green products is more socially recognized than that for self-interested green products [20], so there is no need to invest too many cognitive resources to make trade-offs. However, with self-interested green products, one needs to balance the relationship between self and environmental interests and, thus, invest more cognitive resources than for other-interested green products. Furthermore, consistent with our prediction H3b, we also found that, under loss framing, green consumption decisions for self- and other-interested green products evoked statistically equivalent P3. As mentioned in the introduction, due to the existence of loss aversion, the individual’s focus was shifted to the situation of loss rather than the psychological distance [31].

Regarding the behavioral results, these were not consistent with our hypotheses H1a and H1b. We did not find an interaction between green product types and message framings. Previous studies have found a gap between green purchase intentions and actual green purchase behaviors [10,48]. Kulkarni and Yuan [25] measured the persuasive effect of advertising using a scale of willingness, while we measured the actual green purchase behavior. This may indicate that the influence of green product types and message framing is reflected in intention rather than actual green purchase behavior. However, we observed that, regardless of the type of green products, loss framing induced a higher proportion of green product purchases than gain framing. Consistent with previous studies, losses can promote green consumption behavior more than gains [60,61]. Since individuals are more sensitive to loss than gain, they make more green product purchase decisions under loss framing because of the psychology of loss aversion. In terms of decision time, we found that it took the participants longer to decide to purchase green products than to purchase common products. This again verifies that the essence of green consumption behavior is a trade-off between self-interest and environmental interest [8]. This is because we linked the green products purchased by the participants in the experimental task to the participants’ costs, and therefore required them to pay for their own behavior. Our experimental paradigm measured participants’ actual green product purchase behavior. Therefore, the participants experienced sufficient trade-offs when choosing to purchase green products, invested more cognitive resources, and showed that their decision times increased. At the same time, we also found that for self-interested green products, the decision-making time required for the participants to purchase green products was longer than that for common products, while for other-interested green products, the decision-making time for participants to purchase green products was similar to that for common products. As we mentioned earlier, our society advocates a socially oriented and socially responsible manner of green consumption, rather than an egocentric one [20]. Protecting the environment out of altruistic motives is more in line with social environmental protection norms than self-interested motives, so one does not need to make too many choices with other-interested green products, but for self-interested green products, decisions need to be weighed more.

Our findings have several theoretical implications. First, the results highlight different neural mechanisms involving the influence of green product types on green consumption behavior under gain and loss framings. To date, the vast majority of research regarding green consumption has focused on effective solutions to promote green consumption willingness. In the current research, we explored effective marketing schemes to promote actual green consumption behaviors and found that, under gain framing, green product types influenced individuals’ cognitive resource input. Under loss framing, the green product types would affect the negative emotions and emotional conflicts of individuals. These findings broaden the previous insights into understanding green consumption behaviors. Second, we can provide insights for green product marketing to a certain extent, which contributes to a better environment, healthier consumers, and a larger market for green marketing for businesses [62]. Loss framing is more effective than gain framing in promoting actual green purchasing behavior, whether it is a self-interested green product or an other-interested one. We have summarized the results of the study and their implications, as detailed in Table 5.

The present findings also suggest some limitations and several directions for future research. First, in our experiments, the interaction between the green product type and the message framing that we expected did not appear in the behavioral results, although there was such an interaction in the EEG results, especially for P260 and P3. ERP results could reflect the individual’s internal neural mechanism and time-processing process [63], showing that the impact of green product type and message framings on green consumption decisions might only be reflected in neural indicators, not in behaviors. The persuasiveness of the message framings may be insufficient to affect actual behavior, which is often accompanied by other variables [64], such as emotion [65]. Subsequent research can consider emotional factors and further investigate whether it can achieve the effect of promoting explicit actual green buying behavior. Second, there is a difference in psychological distance between self-interested green products and other-interested green products [6], although we did not measure their use scales in formal experiments. Subsequent research should fully consider whether psychological distance is a robust key variable between these two types of green products. Third, this study noticed the existence of the “attitude–behavior” gap, but only examined the actual green purchase behavior in the experiment and did not measure the purchase intention of the participants for green products. Therefore, the existence of this gap can be measured and investigated simultaneously in subsequent studies. Fourth, although the experimental setting of this study controlled additional variables as much as possible, we were unable to examine and exclude all factors related to green consumption. We mainly focused on the effects of green product types and message framing on green consumption, but we did not take into account the personality characteristics of the participants. Previous studies have found that social value orientation will affect how individuals engage in green consumption behaviors. The higher the prosocial value orientation of individuals, the more inclined people are to engage in green consumption behaviors [66]. This personality trait is also an important variable that can be explored in subsequent experiments. In addition, in the selection of the subject group, we only recruited college students and did not pay attention to social people. Subsequent research can expand the selection range of the participant group to continue to investigate the role of green product types and message framing in green purchasing behavior. Fifth, there may be cultural differences between China and the West. Our research was conducted in the context of Chinese collectivist culture, which emphasizes the role of social collectives, while Western culture emphasizes individualism [67]. Choices may be different under these cultural differences. Subsequent cross-cultural research can be conducted to observe whether different green product types have different green product purchasing behaviors under different message framings.

## 5. Conclusions

To sum up, this study explores the effects of green product types and message framing on green purchasing behavior through an ERP approach. The main findings are as follows: First, loss framing can effectively promote consumers’ green purchasing behavior, which is reflected in the higher purchase rate of green products under loss framing compared with gain framing. Secondly, when purchasing self- (vs. other-) interested green products, purchasing green products requires more cognitive resources and trade-offs than purchasing common products, which manifests in longer decision-making time for purchasing green products under the self-interested green products condition than that for common products. Third, under loss framing, consumers have stronger negative emotions and conflicts towards other-interested green products than towards self-interested green products. This is reflected in the fact that other-interested green products have a larger P260 amplitude compared with self-interested green products. Fourth, under gain framing, consumers weigh less and invest fewer cognitive resources in other-interested green products than self-interested green products, which means that other-interested green products elicit a smaller P3 amplitude than self-interested green products.

## Figures and Tables

**Figure 1 ijerph-19-10746-f001:**
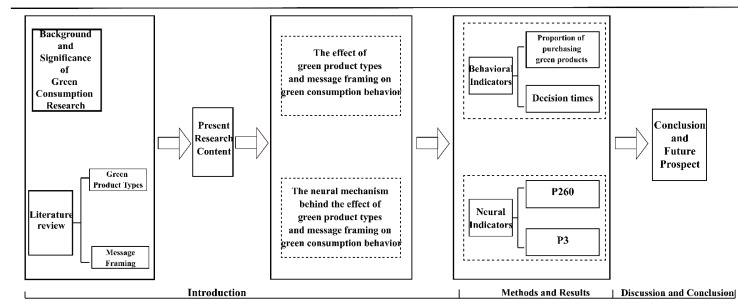
A technical roadmap of the article.

**Figure 2 ijerph-19-10746-f002:**
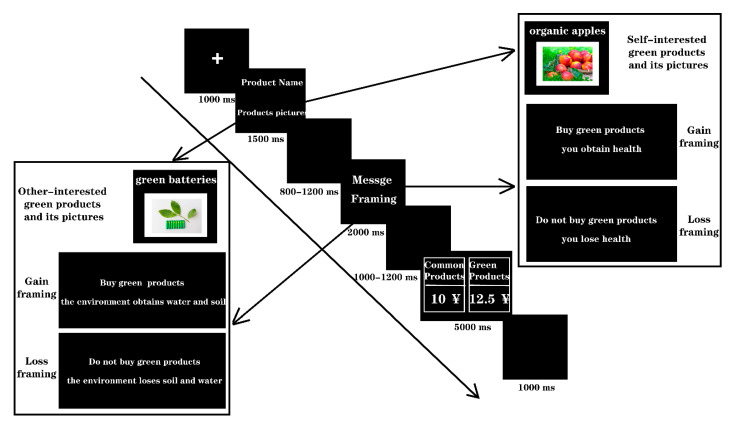
Illustration of a single trial of the experimental procedure.

**Figure 3 ijerph-19-10746-f003:**
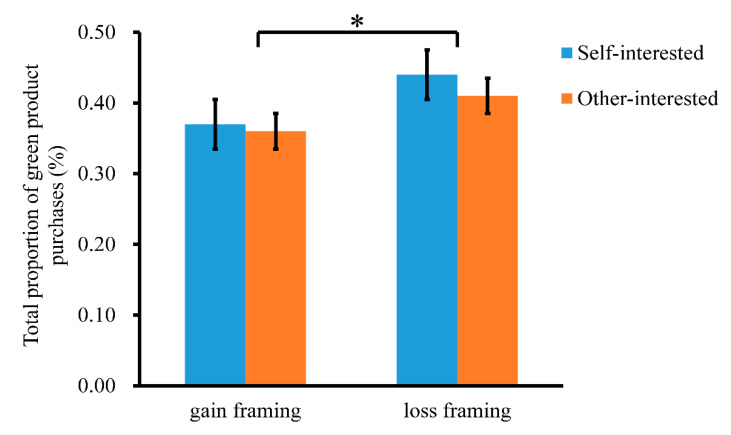
Bar graph of the total proportion of green product purchases in each condition. * *p* < 0.05. Self-interested: self-interested green products, Other-interested: other-interested green products.

**Figure 4 ijerph-19-10746-f004:**
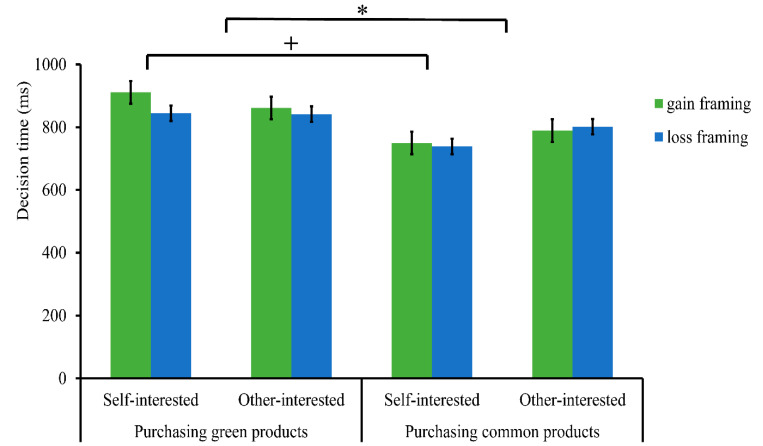
Bar graph of the decision time in each condition. * *p* < 0.05, ^+^ *p* < 0.10. Self-interested: self-interested green products, Other-interested: other-interested green products.

**Figure 5 ijerph-19-10746-f005:**
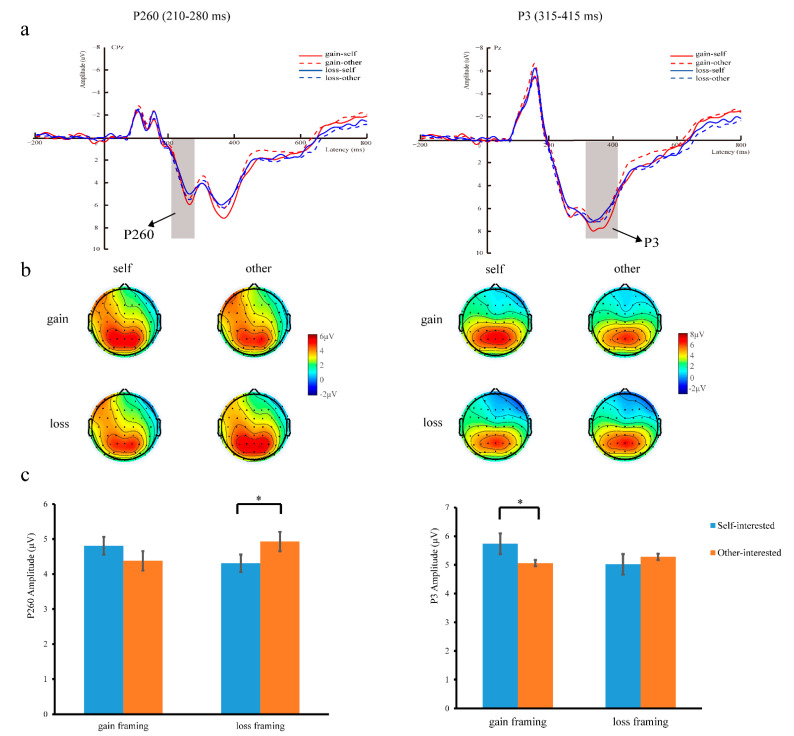
(**a**) Grand-average ERP waveforms at CPz (left) and Pz (right). The grey bars highlight the time window of P260 (210–280 ms) (left) and P3 (315–415 ms) (right); (**b**) Topographies’ voltage distribution of P260 (left) and P3 (right) for each condition; (**c**) The bar graphs of mean P260 and P3 values for each condition. * *p* < 0.05. Self-interested: self-interested green products, Other-interested: other-interested green products. Gain-self: gain framing-self-interested green products, Loss-self: loss framing-self-interested green products, Gain-other: gain framing-other-interested green products, Loss-other: loss framing-other-interested green products.

**Table 1 ijerph-19-10746-t001:** Definition of variable and their characteristics.

Variables	Classification	Definition	Example	Features
Green product type	Self-interested	Products for which consumers make purchasing decisions based on self-interested motives, such as health concerns or cost savings	e.g., organic apple	Self-interested motive, healthy, save costs, close distance
	Other-interested	Products for which consumers make purchasing decisions based on altruistic motives to improve environmental resources	e.g., degradable garbage bags	Altruistic motivation, protect environment, save resources, far distance
Message framing	Gain framing	Emphasize what kind of positive results doing something will bring [21]	e.g., if you buy green products, you will gain health (or environment will gain soil and water resources)	Benefit, positive outcome
	Loss framing	Emphasize the negative consequences of not doing something [21]	e.g., if you don’t buy green products, you will lose your health (or the environment will lose water and soil resources)	Loss, negative outcome

**Table 2 ijerph-19-10746-t002:** A review of medical methods used for marketing research.

Researcher	Description of the Method	Measurement of Purchase Decision (Intention or Behavior)	Main Results
Zubair, et al. [37]	Using ERPs approach to explore the role of message framing and self-conscious emotions in green consumption marketing	Green purchase intention (ask participants “Yes or No” whether to buy green products)	1. Participants have higher green purchase intentions in the gain framing than in the loss framing. 2. The loss framing induces greater P2 and LPP than the gain framing.
Zubair, et al. [38]	Using ERPs approach to explore the role of message framing in green consumption marketing	Green purchase intention (ask participants “Yes or No” whether to buy green products)	1. Participants had higher green purchase intentions in the gain framing than in the loss framing. 2. The gain framing induced greater N170 and P3 than the loss framing.
Jing, et al. [48]	Using ERPs approach to explore the role of empathy with nature in green consumer marketing	Green purchasing behavior (set the price of green products to be 15%, 20%, 25%, 30%, 35%, 40% higher than the cost of common products)	1. When purchasing green products, induced natural empathy induced smaller N2 and N400 than the control group. 2. For the natural empathy group, purchasing green products produced smaller N2 magnitudes than purchasing common products.

**Table 3 ijerph-19-10746-t003:** Proportion of purchasing green products in purchasing decisions under four conditions (*M* ± *SD*).

Green Product Type	Message Framing	Total Proportion of Purchasing Green Products (%)
Self-interested	Gain framing	0.37 ± 0.26
	Loss framing	0.44 ± 0.31
Other-interested	Gain framing	0.36 ± 0.25
	Loss framing	0.41 ± 0.29

**Table 4 ijerph-19-10746-t004:** Decision time in purchasing green products and common products under four conditions (*M* ± *SD*).

Purchase Decision Response	Green Product Type	Message Framing	RT (ms)
Common products	Self-interested	Gain framing	749.96 ± 364.19
		Loss framing	738.49 ± 405.53
	Other-interested	Gain framing	789.06 ± 374.16
		Loss framing	801.62 ± 377.51
Green products	Self-interested	Gain framing	911.46 ± 342.62
		Loss framing	844.44 ± 364.21
	Other-interested	Gain framing	861.27 ± 354.90
		Loss framing	841.77 ± 394.48

**Table 5 ijerph-19-10746-t005:** A table summarizing the results and implications.

		Results	Implications
Behavior results	Proportion of purchasing green products	Participants preferred to purchase green products under loss framing more than under gain framing.	Loss framing is more effective than gain framing in promoting actual green purchasing behavior.
	Decision time	Participants spent more time in deciding to purchase green products than common products.	It is verified that there is a dilemma and conflict in green consumption behavior.
		Compared with other-interested green products, when making purchase decisions for self-interested green products, the decision to purchase green products took more time than common products.	Compared with other-interested green products, individuals face a stronger conflict between egoistic and altruistic motives in self-interested green products.
ERP results	P260	P260 was more positive for other-interested green products than for self-interested green products under loss framing.	Under loss framing, other-interested green products have stronger negative emotions than self-interested green products.
	P3	P3 was less positive for other-interested than for self-interested green products under gain framing.	Under gain framing, other-interested green products have fewer trade-offs and cognitive resources than self-interested green products.

## Data Availability

The data presented in this study are available on request from the corresponding author. The data are not publicly available due to concerns about privacy and ethics in personal decision-making.

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
