# Peer review of "Green Product Types Modulate Green Consumption in the Gain and Loss Framings: An Event-Related Potential Study"

_ijerph, 2022, doi:10.3390/ijerph191710746_

Round 1
Reviewer 1 Report
The differences between self-interested and other-interested green products are not merely psychological distance from individuals. And considering that I did not see any measurements on psychological distance in this paper, this concept should be used with great caution to build the logical framework of paper and interpret results. I think some discussion of results based on psychological distance is appropriate.
How was the impact of costs on green consumption decisions controlled by setting different types of price–cost differences?
For data analyses of RTs, it is recommended to divide trials into two response categories based on participants’ product purchases, since that the proportion of green product purchases were significantly influenced by message framing.
Reviewer 2 Report
Manuscript ID ijerph-1829768
I would recommend a risky revision for this paper. I'm not convinced that this paper fulfil the requirement of this journal, so I will leave this to the editor to decide.
The authors need to make overwhelmingly significant changes to the paper. I hope my suggestions help you do that. Please do not take my comments personally, and good luck! I wish the authors the best in their research endeavors.
Introduction:
*The first paragraph is too limited and only two sentences long. It is suggested to strengthen the expression of the first paragraph because of the lack of robust reality background.
*The paper is technically sound, and the subject matter is presented in a comprehensive manner, but the research motivation is weak, in my opinion. Authors need to link it more robust with the research gaps the article aims at filling in.
*The study should consider including the research questions it seeks to answer in the introduction section so that the process of research can be objectively examined for its directions and methodology.
Materials and Methods
*To improve their literature review, the author(s) should add more recent references related to the topic.
*Figure 1. Some fonts are too small for readability.
Discussions and Conclusions
*The paper lacks robust and coherent arguments on the themes. The paper should engage critically with the previous literature and findings. Thus, discussions and conclusions need to be highlighted more to bring out the study's contribution.
Reviewer 3 Report
- The Abstract lacks research background on green products or green consumption. And the author did not mention the theoretical basis and research methods of this manuscript when describing the main content of the research.
- The author mainly introduces the research results of this manuscript in the Abstract, but lacks a more general and easy-to-understand language to summarize conclusions and contributions of the research.
- In the Introduction, this manuscript does not explicitly mention the innovations of this study and the scientific questions to be solved.
- After reading the Introduction, the research process or research scheme is still not clearly explained in the manuscript.
- Lack of theoretical support in the manuscript.
- In 2.1 Participants and experimental design, the author lacks the support of authoritative literature for the selection of participants and the process of experimental design.
- The conclusion is a bit brief and lacks a general summary.
- The manuscript lacks theoretical or managerial implications, and does not mention the limitations and future prospects of this study.
Reviewer 4 Report
The authors present interesting research merging topics addressed by different scientific disciplines, especially in the methodological part. Limitations of the study are well addressed, also including cultural differences. The text is written in good English, with good general flow. However, the paper addresses definitions and concepts from really different disciplines (economics and business studies/marketing vs. medicine. That this is why I would like to suggest including tables, figures in order to present the content in a way more friendly to readers. For example It will be more clear and understandable if authors present a kind of table/matrix with the classification for self-interested, other interested, gain and loss framing etc. and the rest of features discussed in the text within line 30-66.
This interdisciplinary approach can be also strengthen by a more developed review of the state of the art regarding other research already done in the same/similar way (similar methodology). For example a table with a review of medical methods used for economic/marketing research. The table should include not only the number for the reference cited at the end of the paper but also names of authors, short description of the method applied, main results, geographical location. I think it can be quite easy to include in the paper.
Results can be also widely discussed, maybe with a kind of graphical model summarizing the results.
Authors refer to their predictions – I strongly recommend to clearly formulate hypotheses of the study at the beginning (number them to present them clearly), then verify them and at the end clearly state about the result of this process (negative, positive verification).
Authors use abbreviations in the abstract which are not clear for the wide audience. I strongly suggest to replace the abbreviations with full names/notions.
Figure 1 – bigger font necessary as it is not clear
Round 2
Reviewer 2 Report
The authors try to improve the manuscript. Finally, the font color should be uniform, and there are still blue and black at the same time
Reviewer 3 Report
Despite the authors' efforts to revise this manuscript, I am still concerned.
1. I found that the authors reviewed some literature in the field of green behavior, but lacked a critical review of the most recent literature published in 2022.
e.g.,
Aldubai, O. A., & Develi, E. Ä°. (2022). Green Marketing and Its Impact on Consumer Buying Behavior. Journal of International Trade, Logistics and Law, 8(1), 162-170.
Asl, R. T., & Khoddami, S. (2022). A Framework for Investigating Green Purchase Behavior with a Focus on Individually Perceived and Contextual Factors. Business Perspectives and Research, 22785337221080505.
Druică, E., Vâlsan, C., & Puiu, A. I. (2022). Voluntary Simplicity and Green Buying Behavior: An Extended Framework. Energies, 15(5), 1889.
Li, X., Dai, J., Li, J., He, J., Liu, X., Huang, Y., & Shen, Q. (2022). Research on the Impact of Enterprise Green Development Behavior: A Meta-Analytic Approach. Behavioral Sciences, 12(2), 35.
etc.
2. To enhance the academic logic of this manuscript, I propose to add a technical roadmap.
3. In Section 2.1, the authors introduce samples but ignore representativeness. Please note that not only the quantity, but also information such as occupation and region is also important information that needs to be demonstrated for the representativeness of the sample. Please fully demonstrate the representativeness of the sample.
To sum up, I recommend that the authors carefully revise this manuscript again based on the above comments. I sincerely look forward to receiving the revised version.
Round 3
Reviewer 3 Report
The authors address my comments. The current version is acceptable.